# Non-invasive super-resolution imaging through dynamic scattering media

Dong Wang [1,2,5], Sujit K. Sahoo [1,3,5], Xiangwen Zhu [1], Giorgio Adamo [4] & Cuong Dang [1✉]

Super-resolution imaging has been revolutionizing technical analysis in various fields from biological to physical sciences. However, many objects are hidden by strongly scattering media such as biological tissues that scramble light paths, create speckle patterns and hinder object's visualization, let alone super-resolution imaging. Here, we demonstrate non-invasive super-resolution imaging through scattering media based on a stochastic optical scattering localization imaging (SOSLI) technique. After capturing multiple speckle patterns of photo-switchable point sources, our computational approach utilizes the speckle correlation property of scattering media to retrieve an image with a 100-nm resolution, an eight-fold enhancement compared to the diffraction limit. More importantly, we demonstrate our SOSLI to do non-invasive super-resolution imaging through not only static scattering media, but also dynamic scattering media with strong decorrelation such as biological tissues. Our approach paves the way to non-invasively visualize various samples behind scattering media at nanometer levels of detail.

[1] Centre for Optoelectronics and Biophotonics (COEB), School of Electrical and Electronic Engineering, The Photonics Institute (TPI), Nanyang Technological University Singapore, Singapore, Singapore. [2] Key Laboratory of Advanced Transducers and Intelligent Control System, Ministry of Education, and Shanxi Province, College of Physics and Optoelectronics, Taiyuan University of Technology, Taiyuan, China. [3] School of Electrical Sciences, Indian Institute of Technology Goa, Goa, India. [4] Centre for Disruptive Photonic Technologies, SPMS, TPI, Nanyang Technological University, Singapore, Singapore. [5]These authors contributed equally: Dong Wang, Sujit K. Sahoo. ✉email: HCDang@ntu.edu.sg

Optical imaging beyond diffraction-limit resolution has enabled incredible tools to advance science and technology from investigations of the interior of biological cells[1,2] to chemical reactions at single-molecule levels[3]. Super-resolution stimulated emission depletion microscopy[4] has progressed rapidly to achieve three-dimensional (3D) imaging with super-high spatiotemporal precision[5,6]. For single-molecule detection and localization approaches[7,8], such as stochastic optical reconstruction microscopy or photo-activated localization microscopy, positions of photo-switchable probes are determined as centers of diffraction-limited spots. Repeating multiple imaging processes, each with a stochastically different subset of active fluorophores, allows nanometer-resolution image reconstruction[9]. After these pioneering techniques, super-resolution microscopy has developed rapidly with various other techniques[10–12] to bring the optical microscopy within the electron microscopy resolution. However, the requirement of sample transparency makes the super-resolution microscopy techniques impossible to access objects, which are hidden by strongly scattering media such as biological tissues, frosted glass, or around rough wall corners (Fig. 1a and Supplementary Figs. 1 and 2). These media do not absorb light significantly; however, they scramble the light path, create noise-like speckle patterns[13], and challenge even our low-resolution visualization of samples.

Many approaches have been demonstrated to overcome the scattering effects and enabled imaging or focusing capability through scattering media. The most straightforward strategies utilize ballistic photons[14–16]. However, strongly scattering media reduce the number of ballistic photons and lower the signal tremendously[17]. Some techniques require a guide star or access to the other side of the scattering media to characterize or reverse their scattering effects before imaging such as wavefront shaping techniques[18–21] or transmission matrix measurement[22,23]. Another approach relies on the memory effect of light through

scattering media[24,25], which implies a shift-invariant point spreading function (PSF). A scattering medium with a known PSF (often being measured invasively) can be treated as a scattering lens for imaging by deconvolution[26–28] (Fig. 1b). Similar any conventional lens, scattering lens can only resolve objects up to diffraction limit defined by its numerical aperture (NA) as illustrated in Supplementary Fig. 2d–f. Deconvolution imaging currently provides the best-resolution images from speckle patterns with minimum media characterization (single-shot PSF measurement). However, each measured PSF is only valid for scattering characteristics at the moment of measurement; therefore, the deconvolution method works well for static scattering media, but it cannot be used practically for dynamic scattering media. Non-invasive imaging through scattering media, where the image is retrieved without any measurement of scattering media, is desired for real applications. Diffuse optical tomography[29,30] and time-of-flight imaging[14,29,31] are possible solutions; however, with a resolution of several orders lower than the optical diffraction limit. Thanks to the shift-invariant speckle-type PSF of thin scattering media, the two-dimensional (2D) image and even the 3D image of a sample can be revealed non-invasively from the speckle patterns by a phase-retrieval algorithm[32–34]. The limited performance of the algorithm and cameras, together with the presence of noise and sample's complexity, usually makes the image retrieval process failed or converged with some artifacts and lower resolution compared to the diffraction limit and certainly to deconvolution images (Fig. 1c and Supplementary Fig. 2g–i).

Here we present our stochastic optical scattering localization imaging (SOSLI) technique to do non-invasive super-resolution imaging through scattering media. The approach only requires an imaging sensor to capture speckle patterns created by blinking point sources behind a scattering medium (Figs. 1a and 2a, and Supplementary Fig. 1). The positions of point sources in each

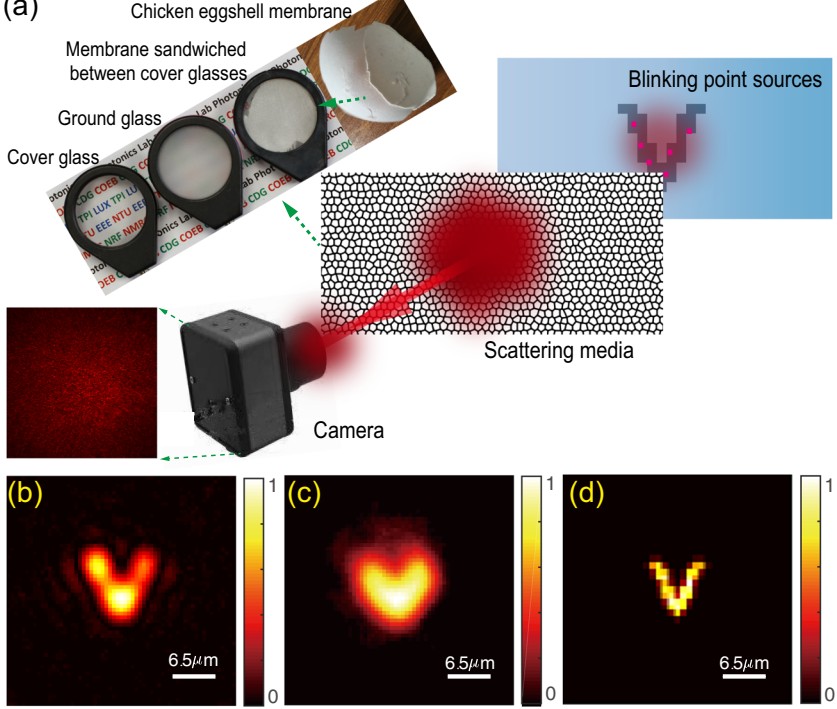

**Fig. 1 Super-resolution imaging through scattering media with SOSLI in comparison to other imaging techniques. a** Schematic of SOSLI where incoherent light from blinking point sources hidden behind various scattering media is scattered and then captured by a camera. **b–d** Experimental demonstrations of the current state-of-art invasive imaging, non-invasive imaging, and our super-resolution SOSLI, respectively, through the same scattering medium in the identical experimental setup.

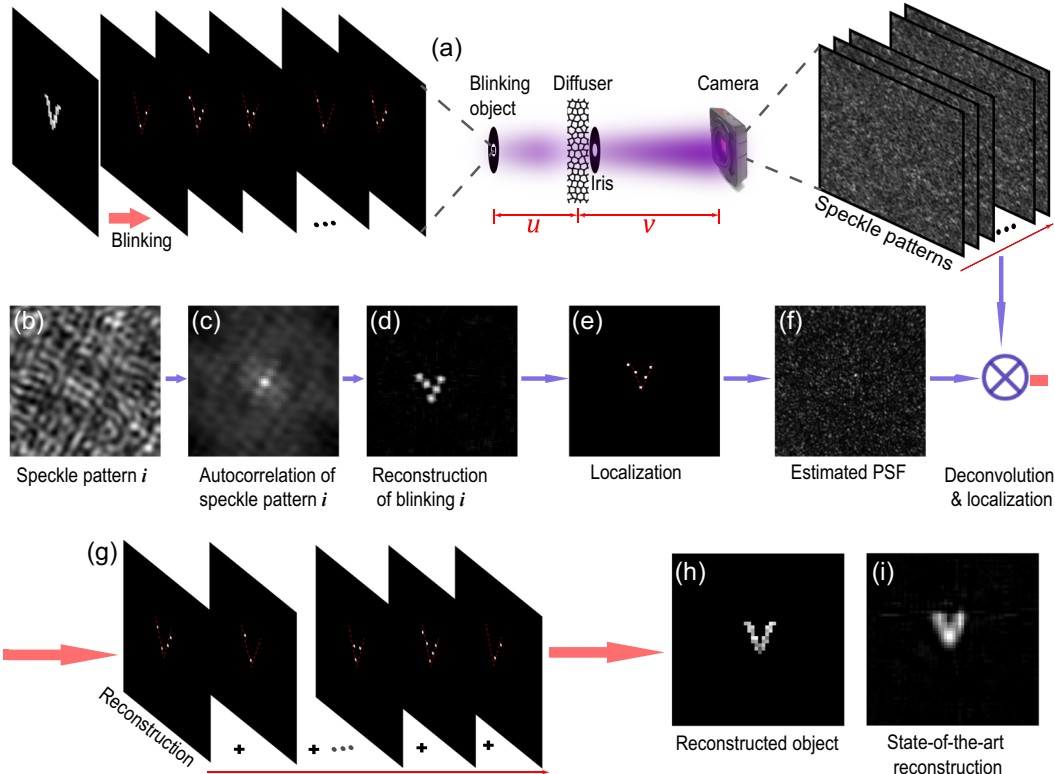

**Fig. 2 Principle and simulation results of SOSLI. a** Object constitutes many intermittent point sources behind an optical diffuser; the iris defines the optical aperture of the imaging system, and the camera captures speckle patterns. **b** A small portion of a typical speckle pattern. **c** Autocorrelation of the speckle pattern is similar to that of the point source pattern. **d** A retrieved image from its autocorrelation. **e** Localized point sources from the retrieved image. **f** Estimated PSF′ from the localized point source frame (**e**) and its corresponding speckle pattern (**b**). **g** A series of localized point source images by deconvolution of the speckle patterns with the estimated PSF′. **h** A reconstructed image with a sub-diffraction-limit resolution by superposing all the individual localized source images. **i** A retrieved image from a single-shot speckle pattern when all point sources are on, i.e., the current state-of-art non-invasive imaging scheme[33].

stochastic frame are determined computationally at very high precision, allowing super-resolution image reconstruction (Fig. 1d). We demonstrate the resolution beyond the diffraction limit by a factor of eight as a proof of concept. Although 100 nm feature size behind the scattering media is well resolved in our demonstration, the resolution limit of our SOSLI is governed by signal-to-noise ratio (SNR), similar to other computational super-resolution microscopy techniques for transparent samples. More interestingly, the localization algorithm is based on a single-shot speckle pattern with minimum scattering media's correlation at the shooting time; therefore, we develop adaptive SOSLI to do super-resolution imaging through dynamic scattering media, such as fresh chicken eggshell membranes where the correlation is as low as 0.20. Our SOSLI demonstrates a desired technique to see through translucent media such as biological tissues or frosted glass with a deep sub-wavelength resolution.

## Results

**Stochastic optical scattering localization imaging**. An object $O$ consists of stochastically blinking point sources: $O = \sum_{i=1}^{N} O_i$, where $O_i$ is the $i^{th}$ blinking pattern (a subset of $O$) and $N$ is the total number of the blinking patterns. After light propagating through scattering media, each $O_i$ produces a speckle pattern $I_i$, captured by a camera (Fig. 2a). If object size is within the memory effect of the scattering media, the PSF is shift-invariant speckles; therefore, the speckle pattern $I_i$ (Fig. 2b) of object $O_i$ preserves the object's autocorrelation[33] (Fig. 2c). The image of $O_i$ can be

retrieved from its autocorrelation by an iterative phase-retrieval algorithm[33] (Fig. 2d). The limitation in the camera's bit depth, photon budget, and performance of phase-retrieval algorithms in the presence of image acquisition noise degrades the diffraction-limit resolution of this non-invasive retrieval image. However, a standard localization algorithm[35,36] is employed to find the position of point sources at a very high resolution (Fig. 2e) and remove algorithm artifacts. Similar to other localization microscopy techniques, the precision is higher for samples with spatially sparse point sources where only one source is temporally active in a diffraction-limited region. The sharp and clear image $O_i'$ presents the precise relative point source positions of pattern $O_i$, while losing their exact positions, because $O_i'$ is only retrieved from autocorrelation of $O_i$ through autocorrelation of $I_i$. The estimated PSF of the scattering medium can be retrieved by deconvolution (Fig. 2f): $PSF' = Deconvol(I_i, O_i')$, which is also shifted in relation to the actual PSF because of the shift in the position of $O_i'$ compared to $O_i$. Besides losing the exact position, phase-retrieval algorithm cannot distinguish the true solution with its flip via a central inversion. Therefore, we deduce different PSF′ corresponding to different flipped versions of $O_i'$, then validate them in deconvolution with another speckle image ($I_j$) to determine the correct one. It is worth noting that the phase-retrieval algorithm performs better′ with sparse samples because of not only the simpler solution but also the higher contrast of the speckle pattern. This phase-retrieval solution will affect the subsequence localization and PSF estimation. We should choose the speckle image with the highest contrast for phase retrieval and PSF estimation[37].

Next, a series of clean super-resolution images $O'_j$ consisting of point source positions is reconstructed for a corresponding series of stochastic patterns $O_j$ by deconvolution of its corresponding speckle pattern $I_j$, with the estimated PSF′ and localization as presented in Fig. 2g. A super-resolution image of the full sample (Fig. 2h) is now reconstructed by superposing all individual images as: $O' = \sum_{j=1}^{N} O'_j$, which represents the object $O$ with an arbitrary position. This principle is valid, provided that PSF does not change among the group of $I_j$. For comparison, we present the typical simulation image (Fig. 2i) retrieved from autocorrelation of a single speckle pattern, in which simulation parameters are similar, with the exception that all the point sources are on. The PSF is generated with a low-pass filter whose cutoff frequency is 1/3.2 times of the Nyquist frequency. Therefore, the simulated diffraction limit is about 3.2 pixels. This current state-of-the-art technique for non-invasive imaging through scattering media shows a blurry image (diffraction limit of the optical system) together with some artifacts from the phase-retrieval algorithm. In contrast, the image reconstructed by SOSLI is much sharper with single-pixel resolution (Fig. 2h).

**Super-resolution imaging through a ground glass diffuser.** To prove our concept, we first demonstrate SOSLI for non-invasive super-resolution imaging through a ground glass diffuser. Microscopic objects comprising multiple stochastic blinking point sources are created by de-magnifying projector images through a microscope objective. The de-magnifying image of each pixel in a digital micro-mirror device (DMD) is an intermittent source with a size of about 1.34 μm (Supplementary Fig. 1a). The microscopic object is placed 10 mm behind the ground glass diffuser, but this distance is kept unknown in all demonstrations. The object's incoherent light propagating through the optical diffuser is recorded by a monochromatic camera, which is 100 mm in front of the diffuser. An iris with a diameter of 1 mm is placed immediately after the optical diffuser to act like the aperture of the imaging system (Supplementary Fig. 1b). A larger iris size enhances the imaging system's diffraction limit and achieves a sharper image (Supplementary Fig. 2); however, it reduces the speckle contrast that is vital for the phase-retrieval approach.

Figure 3 shows the experimental measurements and results for three different imaging approaches. Single-shot non-invasive imaging through scattering media is performed in Fig. 3a–c, where all point sources are on. The image is recovered from the autocorrelation of a single speckle pattern (Fig. 3a) by the phase-retrieval algorithm[33]. The image is very blurred and we cannot distinguish two lines with a gap of 4 μm between them (Fig. 3b, c). This is understandable if we calculate the diffraction

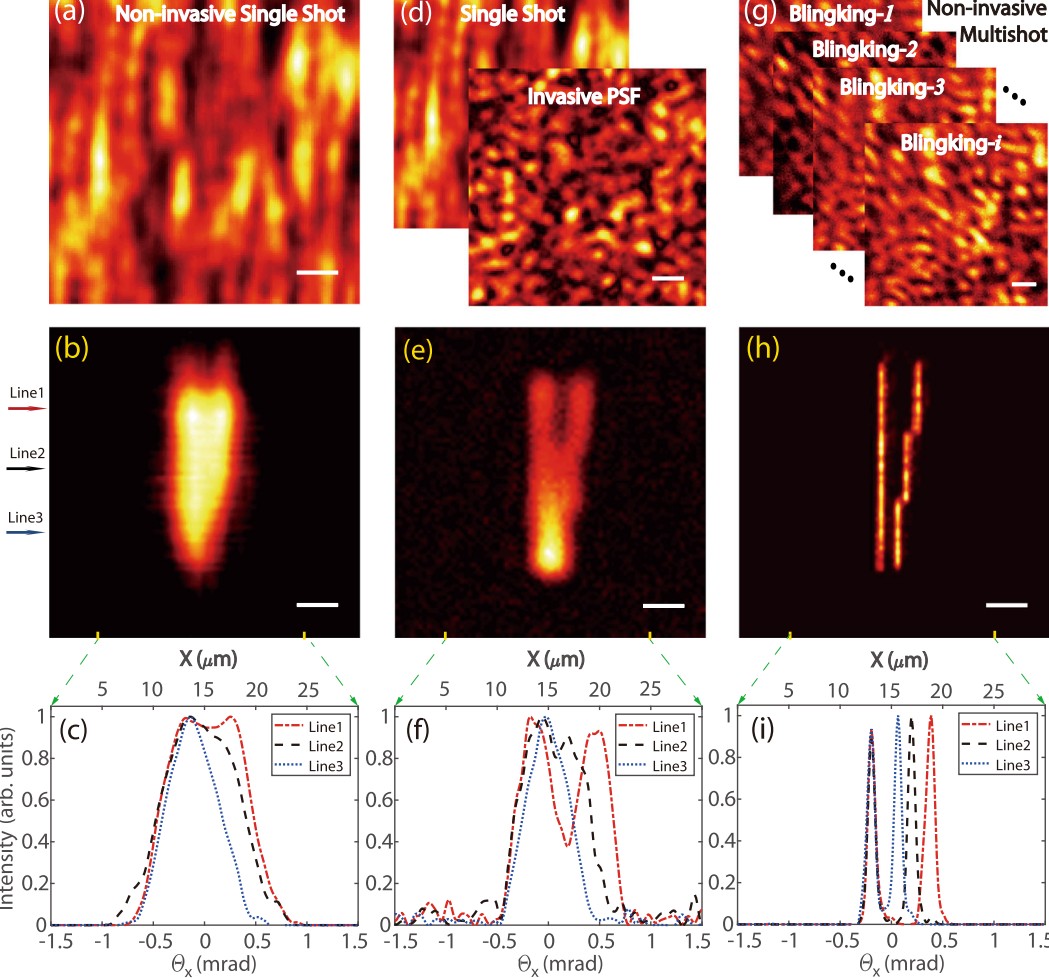

**Fig. 3 Experimental results of imaging through a ground glass diffuser with different techniques. a** Non-invasive single-shot speckle pattern for phase-retrieval algorithm. **b, c** Phase-retrieved image and its intensity profile, respectively. **d** Invasively measured PSF and single-shot speckle pattern for deconvolution imaging. **e, f** Deconvolution image and its intensity profile, respectively. **g** Non-invasive speckle patterns from a stochastic sample for SOSLI. **h, i** Non-invasive super-resolution imaging by our SOSLI and its intensity profile, respectively. Three arrows on the left indicate the three lines for cross-sectional intensity curves in **c, f, i**. Scale bars: 10 camera pixels, equivalent to 6.5 μm on the object plane.

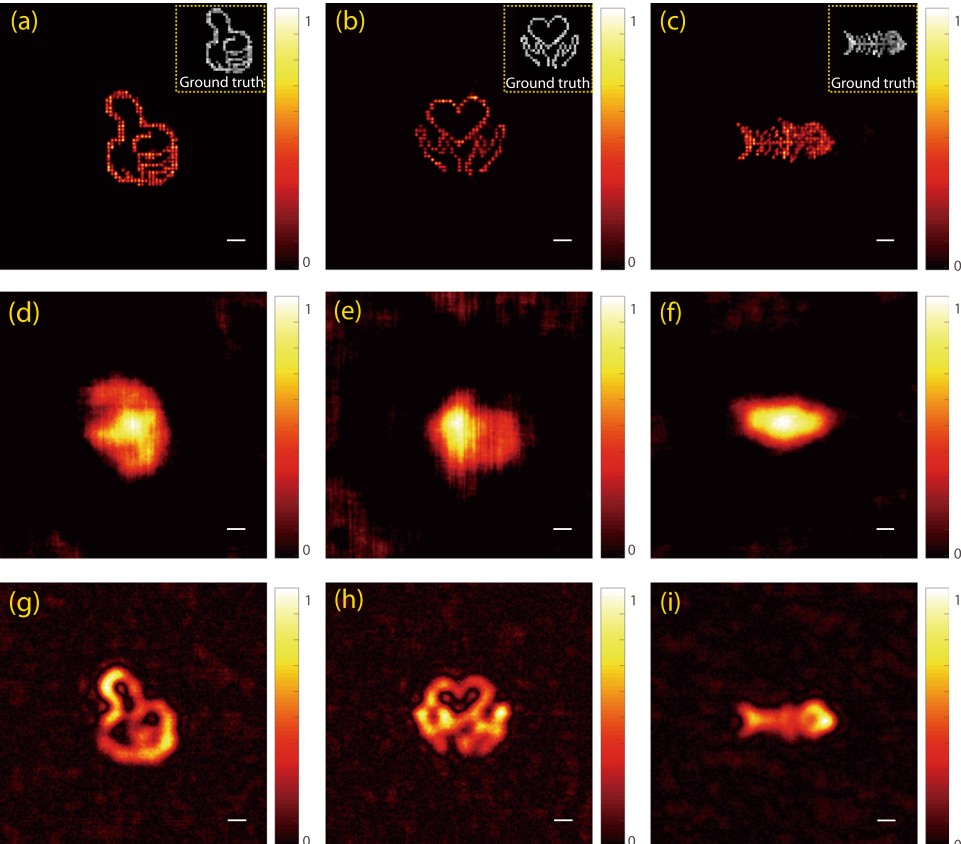

**Fig. 4 Experimental demonstration of three techniques for imaging several complex objects hidden behind a ground glass diffuser. a–c** Our SOSLI approach for non-invasive super-resolution imaging. Insets are ground truth objects. **d–f** Non-invasive imaging retrieved from autocorrelation of a single speckle pattern for the ground truth samples in the insets of **a**, **b**, **c**, respectively. **g–i** Invasive imaging by deconvolution approach with an invasively measured PSF for the ground truth samples in the insets of **a**, **b**, **c**, respectively. Scale bars: 6.5 μm.

limit of our system as $0.61\lambda/\text{NA} = 6.7$ μm, where $\lambda = 550$ nm and $\text{NA} = 0.05$. Beside the diffraction limit, the performance of the phase-retrieval algorithm in the presence of experimental noise degrades the image quality and limits the resolution. With the DMD projector, we can measure the PSF by turning on a single pixel at the center only and capture its speckle pattern (Fig. 3d). Such an "invasive guiding star" for the PSF measurement allows us to calculate the image by deconvolution and significantly enhances the resolution (Fig. 3e). The invasive deconvolution approach is more deterministic, robust to the noise, and enhances the image's high spatial frequency components. We are now able to distinguish the two lines with a 4 μm gap but still cannot see a 2.68 μm gap (Fig. 3e, f). Most strikingly, our super-resolution image reconstructed non-invasively by SOSLI is remarkably clear as presented in Fig. 3g–i. We can resolve very well all the smallest features of our sample, i.e., two thin lines (1.34 μm width) with a gap of 1.34 μm in between (Fig. 3h, i). Due to the present limitation of the projector's pixel size and optics of the sample creating system (Supplementary Fig. 1a), these smallest sample features are only smaller than the diffraction limit by a factor of 5. However, the high contrast between the lines and the narrowest gap (Fig. 3h, i) clearly illustrates that the capability of our demonstrated SOSLI is far beyond 1.34 μm resolution.

It is worth highlighting some important factors in our SOSLI performance. Supplementary Fig. 3 presents more detail for the reconstruction process in which the localization is essential in removing all the background noise and artifacts resulting from the phase-retrieval algorithm. This localization leads to a better estimation of PSF for a series of deconvolution calculation after

that. Better PSF estimation indeed increases the precision of deconvolution and subsequent localization, but SOSLI seems to tolerate more errors in PSF estimation (Supplementary Figs. 4 and 5). Our approach relies on stochastic patterns of point sources to reconstruct a full object; therefore, the image quality is improved with more stochastic patterns (Supplementary Fig. 6). Figure 4 presents some images of more complex samples for performance comparison among the three techniques. Similar to Fig. 3, the complex objects are best resolved with our SOSLI approach (Fig. 4a–c), whereas the retrieval image from auto-correlation of a single speckle pattern shows the poorest performance with some artifacts (Fig. 4d–f). The invasive imaging approach by deconvolution offers moderate performance in Fig. 4g–i. Obviously, our SOSLI for non-invasive imaging through scattering media goes far beyond the diffraction limit and surpasses all the current state-of-the-art imaging through scattering media, including both invasive and non-invasive techniques.

**Super-resolution imaging through a biological tissue.** Our SOSLI demonstrations in Figs. 3 and 4 rely on a fixed PSF for the reconstruction of multiple stochastic patterns and, therefore, we cannot directly use for dynamic scattering media such as biological tissues. Figure 5a shows the decorrelation behaviors of PSFs for two different scattering media. For static scattering media such as a ground glass diffuser, the PSF is a constant pattern and the correlation of 1 is achieved for any measured PSFs at any time. On the other hand, dynamic scattering media such as fresh

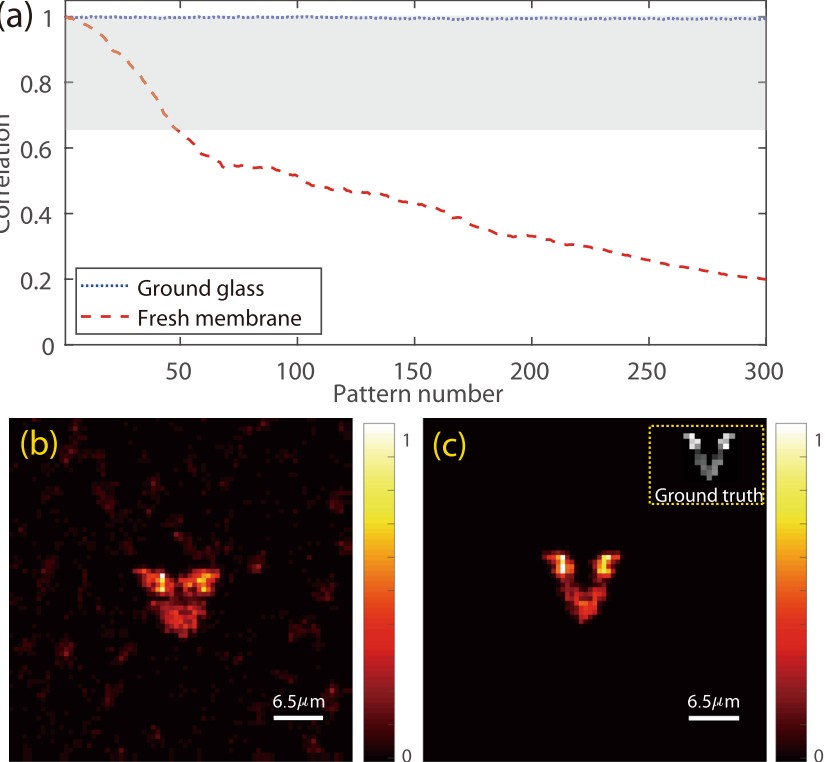

**Fig. 5 Experimental demonstration of non-invasive super-resolution imaging through a fresh chicken eggshell membrane by SOSLI. a** Speckle correlation of PSFs at different measurement time for the static scattering medium (ground glass) and the dynamic one (the fresh chicken eggshell membrane). **b** The reconstructed image by SOSLI with a single estimated PSF. **c** The reconstructed image by SOSLI with adaptive PSF estimation.

chicken eggshell membranes, the PSF gradually changes and the correlation with the initial one decreases with time. In our experiment for the fresh chicken eggshell membrane, the correlation reduces from 1.0 to 0.20 after 300 measurements, with the fastest decay rate in the first 70 measurements (correlation decreases to 0.54). For SOSLI, we also measure 300 stochastic patterns in the same condition, so the membrane's decorrelation behavior is expected to be as shown in Fig. 5a. The reconstruction by SOSLI with a single estimated PSF shows a noisy and blurred image (Fig. 5b) due to this reduced correlation (from 1.0 to 0.20) of the membrane. Supplementary Fig. 7 presents the deconvolution images from several speckle patterns with the PSF estimated from the first speckle pattern. Certainly, the assumption of a static PSF in SOSLI does not hold in this case.

We introduce an adaptive approach to demonstrate our SOSLI for super-resolution imaging through dynamic scattering media. As presented in Supplementary Fig. 8, we now utilize SOSLI to localize point sources in a group of first 50 stochastic patterns, in which the fresh chicken eggshell membrane still can retain its PSF correlation of >0.65 (Fig. 5a). From the last speckle patterns of the group and its corresponding localized point sources, we re-estimate the PSF. This newly estimated PSF is used to reconstruct a new group of the next 50 speckle patterns. The process is continued for all 300 collected stochastic speckle patterns (6 groups) to complete a SOSLI image as presented in Fig. 5c. The final image with adaptive SOSLI is super-resolution, much clearer and less noise than SOSLI with a static PSF (Fig. 5b). For comparison, the single-shot images through the chicken eggshell membrane obtained by the phase-retrieval algorithm and invasive deconvolution are similar to Fig. 1b, c, respectively. The adaptive SOSLI mitigates the decorrelation problems of dynamic scattering media and allows us to reconstruct a super-resolution image (Fig. 5c) with an effective correlation of >0.65 (shaded area in

Fig. 5a), whereas the actual correlation reduces to 0.20 during image acquisition. Certainly, the dynamic scattering media degrade image reconstruction quality compared to that of static media due to the artifact in deconvolution using slightly decorrelated PSF. However, its resolution is still better than single-shot invasive and non-invasive imaging (Fig. 1b, c). The procedure can continue with more stochastic speckle pattern acquisition and the membrane is even completely decorrelated, but the effective correlation for adaptive SOSLI still maintain at >0.65.

We can further modify the process to be the most adaptive SOSLI for highly dynamic scattering media where the deconvolution might fail even when using the estimated PSF for the immediately following speckle pattern. Here we can reconstruct the super-resolution image by retrieving every point source pattern from its speckle autocorrelation with the phase-retrieval algorithm and localization. The deconvolution image (with the PSF estimated from the previous speckle pattern) is only used as the initial guess for the phase-retrieval algorithm. This initial guess is crucial to maintain no-shifting and no-flipping among all retrieved images for superposition. In our demonstration, the correlation of only 0.20 for the scattering media between two shots is sufficient to achieve no-shifting and no-flipping in their phase-retrieved images. However, the phase-retrieval algorithm is known for its sensitivity to input parameters and initial guess; although it is rare, we see small shifting but flipping. Phase-retrieval algorithm can clean and correct various artifacts in the deconvolution image (initial guess); this helps in localization but is blind to position errors. Further, certainly, the correlation of 0.20 is enough to identify the flipping direction in the initial guess; therefore, the phase-retrieval algorithm does not change it. To further increase the resolution of our most adaptive SOSLI, we can check the relative position of the estimated PSF with previous

one to shift the retrieved point source frame accordingly before superposing for the super-resolution image (Supplementary Fig. 9).

We would like to note that the deconvolution and localization algorithms are faster than the phase-retrieval algorithm by orders of magnitude. If we want to save the post-processing time, we should utilize the adaptive SOSLI with the most acceptable number of speckle patterns in a group. In many applications, the decorrelation time is unknown, the number of speckle patterns per group needs to be decided non-invasively by calculating the correlation of several estimated PSFs from individual speckle patterns at different time intervals. If post-processing time is not a concern, we can always use the most adaptive SOSLI approach for all the scattering media, whose correlation in two consecutive imaging shots is not <0.20.

## Discussion

We would like to highlight the role of SNR on the resolution, the fundamental factor for all computational super-resolution microscopy, including our SOSLI. For conventional super-resolution microscopy with microscope objectives and transparent samples[7–9], all photons from a single source go to the camera and form its diffraction limit spot, which can be few 10s to 1000s of pixels depending on magnification and NA. However, with scattering media, the photons are scattered everywhere, forming speckles inside and even outside the camera; in addition, we need multiple speckles to retrieve an image. The SNR will be very low in SOSLI for applications with blinking emitters, because SOSLI needs only one point source to be active within the area of diffraction limit in each frame. We might not improve SNR simply by increasing the frame's integration time, as there will be more chances for multiple point sources to be active within diffraction limit area. Besides the need for a low noise and high quantum efficiency camera to enhance SNR, high photon budget (number of photons per blink per emitter) is a challenging requirement. Ultrabright single fluorophores or quantum emitters could emit at the best $10^4$–$10^6$ photons per blink to a camera[38]. With 10–1000 pixels in a diffraction-limited region for standard imaging with an objective lens and transparent samples, this is equivalent to $10^1$–$10^3$ photons per useful pixel (PPP). However, for scattering media, these photons are spread in million pixels; this implies much $<10^{-2}$–$10^0$ PPP, lower than the camera's noise floor. The low photon budget also implies the uncertainty of the captured speckle patterns regardless of noise (Supplementary Fig. 10). Such quantum uncertainty is the significant noise source for SOSLI with intermittent emitters. Our simulation (Supplementary Fig. 11) for static scattering media and noiseless experiments shows that 11% of success rate for phase retrieval and localization can be achieved with about 1 PPP. After PSF is estimated, the success rate for deconvolution and localization is very high such as 68% for 0.1 PPP and 100% for 0.45 PPP and above (Supplementary Fig. 12). Understandably, our bottleneck is the phase-retrieval algorithm; and multi-million photons are required for SOSLI. The future development of inorganic quantum emitters with very high quantum yield (for brightness) and suppressed blinking rate[39] (for a longer exposure time of each stochastic image) could enable SOSLI in practical applications.

Besides the SNR, other main factors affecting SOSLI's performance are the memory effect region, the sample sparsity, and light sources' spectral linewidth. The sample size should be within the memory effect region, which is inversely proportional to the scattering media thickness. The ratio ($R = L/l$) between media thickness ($L$) and the mean free path (MFP, $l$), which is either scattering MFP ($l_s$) for scattering regime or transport MFP ($l_t$) for diffusion regime, typically determines how strong the scattering

effect is. However, the ratio ($R$) does not define how large the memory effect region is. SOSLI can be successful with a very large ratio $R$ (i.e., multiple scattering) if $L$ is small (i.e., the MFP is also small). However, SOSLI can easily fail with a small ratio $R$ (i.e., a few or single scattering) if the media is thick (i.e., both $L$ and $l$ are large). The chicken eggshell membranes (thickness of about 90 μm)[40] with 3D volumetric scatterers are very strong scattering media whose ratio $R$ is 7.3 and 3.6 for scattering and transport MFP ($l_s$ and $l_t$), respectively (more detail in Supplementary Figs. 13–15). Our computational approach starts with the phase-retrieval algorithm, which is known for random artifacts and, therefore, it is advised to try several times to get the best results. Indeed, the sample sparsity (less unknown inputs) together with a large number of speckles per pattern (more measured signals) improves the autocorrelation estimation and enhances the phase-retrieval algorithm's performance. For the last point, SOSLI relies on incoherent light speckle patterns, which are low contrast if the spectral bandwidth is large. Our 10 nm bandpass filter certainly helps to improve the speckle contrast from the projector's pixels. Several inorganic quantum emitters (such as quantum dots, nanoplatelets) can provide such narrow spectral bandwidth for potential demonstrations.

Demonstrating SOSLI at nanometer resolution is desired but requires nanoscale intermittent point sources with a high photon budget. Our demonstration with de-magnifying DMD pixels allows us to illustrate SOSLI closest to the actual practice, but we must intentionally reduce the NA of the imaging system to show the super-resolution effect for 1.34 μm feature size. We now demonstrate SOSLI with a three-bright-point sample, which is about 1 mm behind the ground glass. The diameter of each point is 100 nm. To increase the signal, scattering photons after scattering media are collected by a microscope objective (NA = 0.6) to a camera. There is no aperture in the setup. The scattering angle of ground glass, the field of view, and the NA of microscope objective define the overall NA of the SOSLI system. A one-dimensional piezo stage moves the three-point sample randomly within 6 μm horizontal distance. The camera captures a speckle pattern at every sample position for SOSLI. The inset of Fig. 6a shows a typical deconvolution image with an estimated PSF. It illustrates the diffraction limit spot of 0.8 μm, implying the overall NA of 0.42 for the SOSLI setup. Figure 6a presents the superposing image of all the diffraction-limited deconvolution images, whereas Fig. 6b shows the SOSLI result, which resolves clearly the 100 nm lines with a 400 nm clean gap between them. The fact that the 100 nm lines appear as single-pixel-wide lines in the SOSLI image, especially in the cross-sectional profile (Fig. 6d), clearly demonstrates our SOSLI resolution limit is 100 nm or smaller.

Our demonstration with random positions of the three-point sample could mimic the stochastic blinking phenomenon of point sources at a certain level, but the relative positions among three points are unchanged in each frame. With the existing data set, we superpose several random speckle patterns to generate new speckle patterns, which would make the speckle patterns of 3, or 6, …, or 18 active points. In each generated speckle pattern, the relative positions among these points become random (inset of Fig. 6c). Such generated speckle patterns can be considered as taken from truly stochastic blinking point sources with very high photon budgets. The reconstructed image by SOSLI from these speckle patterns is super-resolution with minimal artifacts at the gap between two thin lines, because there are possibilities that two points within a diffraction limit spot are on simultaneously. Figure 6d presents the intensity profile across the vertical line of the images; SOSLI can resolve the lines at single-pixel precision. Compared to the diffraction limit, our SOSLI achieves an eight-fold enhancement, which is again currently limited by 100 nm

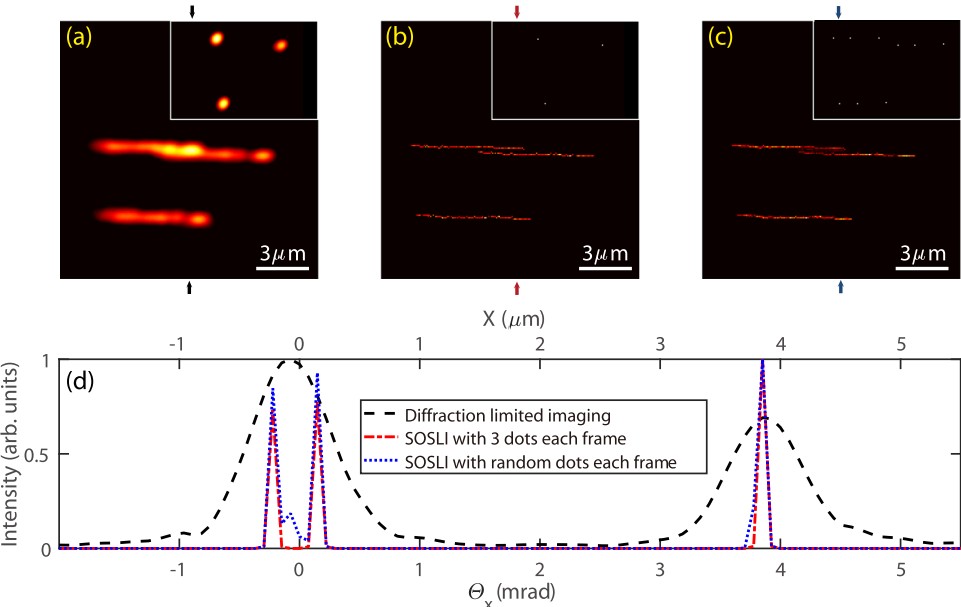

**Fig. 6 SOSLI demonstration with sub-wavelength resolution. a** Superposing multiple low-resolution images of a three-point sample at various random positions; inset: a typical low-resolution image. **b** The SOSLI result of a three-point sample at different random places; inset: a typical localization image. **c** The SOSLI image with randomly appearing points; inset: a typical localization image. **d** Intensity profile across the vertical line of three images (corresponding to the colored arrows on **a**, **b**, **c**).

holes in sample fabrication. The image reconstruction process for SOSLI is illustrated in the Supplementary Video 1.

In summary, we have demonstrated SOSLI for non-invasive super-resolution imaging through both static (ground glass) and highly dynamic scattering media (fresh chicken eggshell membrane). The camera captures multiple images of scattered light from stochastic point sources behind scattering media and then our computational approach localizes these sources non-invasively to reconstruct a super-resolution image. Our experimental results show that SOSLI enhances resolution by a factor of 8 compared to the diffraction limit, resolving 100 nm features with considerably more detail compared to both state-of-the-art invasive and non-invasive imaging through scattering media. Similar to other computational super-resolution techniques, SOSLI's resolution is dictated by SNR which, however, requires a very high photon budget to have acceptable SOSLI performance. The adaptive SOSLI allows super-resolution imaging non-invasively through highly dynamic scattering media with the correlation between two consecutive speckle patterns as low as 0.20. Our SOSLI demonstration shows a promising approach for optical imaging through dynamic turbid media, such as biological tissue, with nanometer resolutions.

## Methods

**Scale bar**. Except Fig. 6, all other experimental results show a scale bar of 10 camera pixels, equivalent to 65 μm in the imaging plane and 6.5 μm on the object plane (the magnification is 10 in our experiments). However, we do not know the scale bar on the object plane or the magnification of the imaging system in the non-invasive approach, because the distance from the object to scattering media is unknown. We can only resolve the sample by angular resolution as presented in Figs. 3 and 6, which is the same for both imaging and object plane.

For the experiment in Fig. 6, the scale bar is calculated based on the fabricated ground truth sample with the exact distance among three points. This approach provides better accuracy than a calculation based on the camera pixel size and imaging optics due to the involvement of microscope objective in the speckle imaging side.

**Scattering media**. The static scattering media are 120 grit ground glass diffusers from Thorlabs or Edmund. The memory effect region is 15 mrad (ref. [27]). The fresh chicken eggshell membrane has a similar memory effect region as that of the glass diffuser; then, the memory effect region decreases when membrane dries.

**Nanoscale sample**. A gold film, 250 nm thick, was deposited on a microscope glass slide. The focus ion beam technique was used to create three 100 nm holes in the gold film. The three-hole gold film was illuminated from behind by a focused green light-emitting diode through an objective lens to create the three-point source sample.

**Data processing**. In all experiments, the resolution of the raw camera images is 2560 × 2160 pixels. We crop them into a size of 2048 × 2048 pixels to implement all the mentioned techniques in this work. The final reconstructed images are cropped to a square window with dimensions ranging between 75 × 75 pixels and 200 × 200 pixels (depending on the imaged object dimensions). Algorithms are developed in Matlab and run on a regular PC (Intel Core i7, 16 GB memory). A typical procedure for SOSLI with 300 speckle patterns takes 2–3 min.

## Data availability

The sample data that support the findings of this study are available in figshare with the identifier https://doi.org/10.6084/m9.figshare.9758669.v1.

## Code availability

The Matlab code in this study are available in figshare with the identifier https://doi.org/10.6084/m9.figshare.9758669.v1.

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

## Acknowledgements

We especially thank Professor Nikolay I. Zheludev at the University of Southampton and Nanyang Technological University, Singapore, together with Professor Sylvain Gigan and his research group at Laboratoire Kastler Brossel, Sorbonne Université, École Normale Supérieure–Paris Sciences et Lettres (PSL) Research University, Paris, France, for great discussions and providing suggestions for improvement. We thank Vinh Tran, Dr. Dayan Li, Dr. Dongliang Tang, and Dr. Huy Lam at NTU Singapore for fruitful discussions and useful feedback. We also thank Dr. Philip Anthony Surman for proofreading. The research is financially supported by the Ministry of Education–Singapore (MOE): MOE-AcRF Tier-1 (MOE2019-T1-002-087), MOE2016-T3-1-006 (S), Nanyang Technological University Singapore (NTU), National Natural Science Foundation of China (grant No. 61805167), and IIT Goa's start-up grant (2019/SG/SKS/014).

## Author contributions

C.D. initiated the idea. S.K.S. developed the Matlab code and performed the numerical simulations. C.D., D.W., and S.K.S. designed the initial experiments. D.W. performed the experiments using a projector with S.K.S.'s participation. X.Z. and C.D. performed the experiments for 100 nm samples. G.A. fabricated 100 nm samples. C.D. and D.W. wrote the manuscript with S.K.S.'s contributions. All authors discussed, analyzed, and took responsibility for the results and revised the manuscript. C.D. supervised and contributed to all aspects of research.

## Competing interests

The authors declare no competing interests.
