## [Peer Review File · Nature Communications]

Reviewers' Comments:

Reviewer #2:

Remarks to the Author:

The authors present a method for non-invasive super-resolution imaging through scattering media. They smartly combined speckle-correlation imaging and stochastic localization microscopy. This paper is well-organized and demonstrations presented here clearly show their achievements. I recommend that this paper will be published in Nature Communications after minor revisions as follows.

1. Exposure times for each experimental demonstration should be described.
2. It is better to add a schematic diagram of adaptive SOSLI for super-resolution imaging through dynamic scattering media in the supplementary materials.
3. Variables "D", "u", and "v" in Fig. 1(a), and "u" and "v" in Fig. 2(a) might not be necessary because they are not referred in the main text. On the other hand, "u" and "v" in Fig. S1(b) should be maintained.
4. Caption at Fig. S3:
...pattern. (i-k) Emitter...
There is no Fig. S3(k).
5. Section 5 in the supplementary materials:
...supplementary Fig. S6a-d after the bicubic interpolation processing. ...
What is the purpose of the bicubic interpolation processing? Please explain.

Reviewer #3:

Remarks to the Author:

The paper entitled "Non-invasive super-resolution imaging through dynamic scattering media" describes an interesting approach to exploit local correlations in speckle patterns to reconstruct sparse objects. Moreover the technique has super-resolution capabilities through localization, which allows it to go beyond the diffraction limit of the imaging optical system. Finally, the manuscript describes how the technique can be adapted to work also with a slowly decorrelating scatterer, which could be fresh biological tissue. However the reader spot some major correction which must be done in order to consider the publication:

1. First, the overall English must be revised, as often ideas are not clear to the reader. I think this is a crucial point for a journal as Nat.Comm
2. Overall statistics must be largely improved regarding the values and figures:
 - a. "but the effective correlation for adaptive SOSLI still can maintain at more than 65%"
 - b. "fresh chicken eggshell membranes with decorrelation of up to 80%"
 - c. fig 5a
 - d. supp fig 10
 - e. supp fig 11
3. Across the manuscript, the terms "Blinking emitters" is used. I think these terms can be misleading, as generally associated with fluorescent emitters which can blink themselves, or after the action of light, as used in many localization-based super-resolution approaches. However here "Blinking emitters are generated by randomly blinking projector pixels", so I suggest to change the term "emitter" to "illumination source", or anything else which takes the reader closer to the actual nature of the system.
4. Parameters of the projection optics are not reported, and so it is not clear to me how the size of

the "blinking emitters" is calculated.

5. "is a scattering lens that, in turn, is a low pass filter like any conventional lens": please make the sentence more clear
6. "However, each measured PSF is only valid for one scattering medium: what the authors mean for medium here? the medium underneath an optical patch? please clarify
7. I suggest to make more explicit how the "100nm resolution" is calculated and the target is made. And please do that close to the first time "100nm" is used across the manuscript.
8. "The simulated diffraction limit is about 3.2 pixels.": please describe briefly how this is calculated
9. "an intermittent emitter with a size of about 1.34 μm (Supplementary Fig. S1a)": please improve the description and the quality of the information provided in supp.fig.1.
10. How the distances "u" and "v" of supp. fig. 1 have been chosen?
11. "We now utilize SOSLI to localize and then superpose emitters in a set of first 50 stochastic patterns": these approach and the numbers of pattern before the evaluation of the PF depends on a prior knowledge of the decorrelation characteristics of the scattering layer, which can be unknowns in most of the applications: I think is needed a discussion of this point.
12. "The SNR is very low with SOSLI and we cannot easily increase it simply by increasing the integration time because we need stochastic nature of blinking emitters from frame to frame": if the authors claim is not possible to have higher SNR with such a simple proof of principle setup, which are the possibilities to move to "real" applications? please discuss this point
13. Data sharing link was not available while revising the manuscript

Some minor points:

14. I suggest to improve the overall quality of Supp.Fig.1: the first two quotes from the left does not have text, I would improve consistency of perspective, resolution of the images (or move from raster to vector graphics), make more clear the light propagation (e.g. a kind of raytrace could be to my view more effective).
15. "transfer matrix measurement" : i think in the field is more often used the term "transmission matrix" instead of "transfer matrix". Unless there is a clear reason to use "transfer", I suggest changing it to be more clear.
16. Another
17. "approach relies on memory effect of light through scattering media and deconvolution imaging": while the phenomenon exploited by the former part of the sentence is memory effect, in the latter the authors refer to the "deconvolution imaging", which now sounds more a technique, not anymore a effect to exploit, (which moreover I assume exploit as well the memory effect/isoplanatism/shift invariance). I think the sentence can bring confusion to the reader, so I suggest to revise it.
18. Report (in response to the referees) as reference of egg shell thickness the site of a mayonnaise producer, without even a datasheet nor the date when the site has been visited, must be not considered a valid response. I suggest finding a better and more scientifically relevant reference.

Although the proposed technique exploits well known techniques, the article describes a simple and effective approach, which has a degree of novelty. Moreover the validation has been with a simple and intuitive setup, which to my view must be considered a plus. However the article still needs to be extensively revised, in order to be clear and the results consistent and corroborated by good statistics. While it could be ready with some minor revision for a more specific journal, I think the quality of the manuscript as presented does not deserve the publication in an high impact and broad audience journal, such as Nature Communication.

Our responses to reviewers' comments

REVIEWER COMMENTS

Reviewer #2 (Remarks to the Author):

Comment. The authors present a method for non-invasive super-resolution imaging through scattering media. They smartly combined speckle-correlation imaging and stochastic localization microscopy. This paper is well-organized and demonstrations presented here clearly show their achievements. I recommend that this paper will be published in Nature Communications after minor revisions as follows.

Reply: We would like to thank reviewer 2 very much for your good words. We are here to address your technical comments point-by-point as below.

Comment 1. Exposure times for each experimental demonstration should be described.

Reply: Exposure time for each demonstration is 1-2 hours. And certainly, it scales linearly with the number of stochastic speckle patterns we would like to capture. The exposure time for each speckle pattern is 5-10 seconds, so we capture about 300 – 700 patterns per hour (accounting for a finite time for the camera to readout, transfer and save files to a PC). We have added this information in the first section of the supplementary material, Optical experiment setup, as below.

The exposure time to capture each speckle pattern is 5-10 seconds to gain good signals.

Comment 2. It is better to add a schematic diagram of adaptive SOSLI for super-resolution imaging through dynamic scattering media in the supplementary materials.

Reply: Yes, we have added new Fig. S8 and its detail description in the supplementary material to show a schematic diagram of adaptive SOSLI. This part is also presented below. Thank you for your suggestion.

7. Adaptive SOSLI for dynamic scattering media

Supplementary Fig. S8. Schematic diagram of the adaptive SOSLI for super-resolution imaging through dynamic scattering media. (a) Multiple groups of speckle patterns. **(b)** Strategy to estimate adaptive PSFs. **(c)** Localized point sources in each speckle pattern.

For dynamic scattering media, a series of speckle patterns are captured with slightly different scattering characteristics. The proposed adaptive SOSLI divides all these speckle patterns into multiple groups (Fig. S8a). The number of stochastic patterns for each group depends on the decorrelation time and image acquisition time. The requirement is that every two adjacent groups have some finite correlation. We do SOSLI with the first group and achieve estimated PSF1 (Fig. S8b) and multiple frames of localized point sources (Fig. S8c). For finite-dynamic scattering media, the decorrelation within a small group of images has not yet affected SOSLI performance. We then use the last speckle pattern and its localized point sources to re-estimate the PSF, which is PSF2 (Fig. S8b). The PSF2 is utilized for deconvolution then localization for all the speckle patterns in the second group (Fig. S8c). The process is continued for all the groups, and a full set of localized-source frames is achieved (Fig. S8c) to reconstruct a super-resolution image by superposition.

Comment 3. Variables “D”, “u”, and “v” in Fig. 1(a), and “u” and “v” in Fig. 2(a) might not be necessary because they are not referred in the main text. On the other hand, “u” and “v” in Fig. S1(b) should be maintained.

Reply: Thank you for this detail suggestion. We agree with this and adjust the figures accordingly in the main text as below.

Fig. 1

Fig. 2

Fig. S1

Comment 4. Caption at Fig. S3:

...pattern. (i-k) Emitter...

There is no Fig. S3(k).

Reply: We appreciate your help here. We must have overlooked this. It should be (i-j) as in the revised version now.

Comment 5. Section 5 in the supplementary materials:

...supplementary Fig. S6a-d after the bicubic interpolation processing. ...

What is the purpose of the bicubic interpolation processing? Please explain.

Reply: Bicubic interpolation processing is used simply to present small images on a larger scale smoother. The original images are small, and we need to enlarge them to present as large images. Bicubic interpolation processing make images less “pixelated”. We add a sentence in the figure caption to explain this point. It is read as follows.

(a-d) The raw results from SOSLI. **(e-h)** The results after bicubic interpolation processing that makes images smoother. Scale bars: 6.5 μm .

Reviewer #3 (Remarks to the Author):

Comment: *The paper entitled “Non-invasive super-resolution imaging through dynamic scattering media” describes an interesting approach to exploit local correlations in speckle patterns to reconstruct sparse objects. Moreover the technique has super-resolution capabilities through localization, which allows it to go beyond the diffraction limit of the imaging optical system. Finally, the manuscript describes how the technique can be adapted to work also with a slowly decorrelating scatterer, which could be fresh biological tissue. However the reader spot some major correction which must be done in order to consider the publication:*

Reply: We would like to thank reviewer 3 for the detailed review report. We are glad to address your comments point-by-point as below.

Comment 1. *First, the overall english must be revised, as often ideas are not clear to the reader. I think this is a crucial point for a journal as Nat.Comm*

Reply: Thank you for your comments. After revising ourselves, we sought support from Dr. Surman, an English scientist doing research on 3D display technology to read and revise language for our manuscript. As a reader outside of the field, Dr. Surman helped us make the manuscript clearer, easier to understand by a broad audience. After thorough revision, we believe the broad readers of Nat. Comm. journal can understand our current manuscript without troubles. As suggested by the editor, we attach the tracked-change version of our manuscript so that reviewers and editor can see all our revision.

Comment 2. *Overall statistics must be largely improved regarding the values and figures:*

a. “but the effective correlation for adaptive SOSLI still can maintain at more than 65%”

b. “fresh chicken eggshell membranes with decorrelation of up to 80%”

c. fig 5a

d. supp fig 10

e. supp fig 11

Reply: We thank you very much for this comment. In the last version, we would like to emphasize how good we can maintain high correlation (>65%) for adaptive SOSLI even when the scattering media is already decorrelated largely (>80%). But we now understand the confusions from readers’ view when we talk about decorrelation and correlation at the same time and the interchangeable use of decimal and percentage numbers (0.65 vs. 65%).

In the revised version, instead of mixing the two concepts, we use only correlation when mentioned about these numbers in the text. Also, instead of using the percentage for correlation, we now use the absolute correlation coefficient, which is from 0 to 1. And we reserve the percentage notation for the success rate in the simulation (Sup. Fig. S11 & S12). In this way, readers can easily distinguish these concepts. The changes are many in various places, you might see them all in our tracked change version.

For Supplementary figures S11 and S12, we have changed the success rate scales to percentage to be consistent with the discussion in the text.

Note that the Supp. Fig. S10 and S11 have been changed to S11 and S12 after adding Fig. S8. These figures are as below.

Fig. S11

Fig. S12

In addition, Decorrelation coefficients in figures S7, S9 have been changed to correlation coefficients as follows.

Fig. S7

Fig. S9

Comment 3. Across the manuscript, the terms “Blinking emitters” is used. I think these terms can be misleading, as generally associated with fluorescent emitters which can blink themselves, or after the action of light, as used in many localization-based super-resolution approaches. However here “Blinking emitters are generated by randomly blinking projector pixels”, so I suggest to change the

term “emitter” to “illumination source”, or anything else which takes the reader closer to the actual nature of the system.

Reply: We understand the reviewer’s point. To make it clearer, we now call this term as “point source” while “emitters” is used a few times in discussion about the practical aspect of the technique.

Thank the reviewer for this comment.

Comment 4. *Parameters of the projection optics are not reported, and so it is not clear to me how the size of the “blinking emitters” is calculated.*

Reply: The exact composition of the projection optics is not known to us either. We utilized a commercial projector from which we removed the projection lens (so that it does not project to a huge screen anymore) while several inaccessible optics (lenses) remain inside. The projector was still able to project a small sharp image in the front near the projector. Then we use an objective (40x, numerical aperture: NA=0.65) to de-magnify and make a much-smaller image for our experiment. As described in Fig. S1, we do not use any other optics, but we still cannot calculate the demagnification factor because of unknown optics inside the projector. We have to measure the size of “blinking emitters”, i.e. the pixel size of our de-magnified screen. We added a paragraph in the first part of the supplementary material to describe our measurement method.

Although the projector’s projection lens is removed so that it does not project to a huge screen, there are still multiple unknown optical components inside as the projector still projects images to the plane of the first iris. Because of these unknown optics, we cannot calculate the demagnification factor from the projector to the object plane. We measure the pixel size of 1.34 μm by a mechanical approach. We turn on 200 pixels to make a few-hundred-micrometer bright line. A blade is then moved along the line to block a portion of it and the transmitted light intensity is monitored by a power meter. We used a differential actuator to control the blade position with sub-micrometer precision. By coordinating the transmission intensity with the blade position, the line length and then the pixel size can be calculated.

Comment 5. *“is a scattering lens that, in turn, is a low pass filter like any conventional lens”: please make the sentence more clear*

Reply: Thank reviewer for this point. We have made our revision in the main text as below.

A scattering medium with a known PSF (often being measured invasively) can be treated as a scattering lens for imaging by deconvolution²⁶⁻²⁸ (Fig. 1b). Like any conventional lens, scattering lens can only resolve objects up to diffraction limit defined by its numerical aperture (NA) as illustrated in supplementary Fig. S2d-f.

Comment 6. *“However, each measured PSF is only valid for one scattering medium: what the authors mean for medium here? the medium underneath an optical patch? please clarify*

Reply: Here “medium” is a singular form of “media.” We, as well as literature, typically talk about scattering media, but here we would like to emphasize on just a single one. To make it easier for readers, we revise this sentence on page 4 as follows.

However, each measured PSF is only valid for scattering characteristics at the moment of measurement; therefore, the deconvolution method works well for static scattering media, but it cannot be used practically for dynamic scattering media.

Comment 7. I suggest to make more explicit how the “100nm resolution” is calculated and the target is made. And please do that close to the first time “100nm” is used across the manuscript.

Reply: Thank you for this important point. We have added a sentence to justify our 100nm resolution claim on page 17 as below.

Figure 6a presents the superposing image of all the diffraction-limited deconvolution images, whereas Fig. 6b shows the SOSLI result, which resolves clearly the 100-nm lines with a 400-nm clean gap between them. The fact that the 100-nm lines appear as single-pixel-wide lines in the SOSLI image, especially in the cross-sectional profile (Fig. 6d) clearly demonstrates our SOSLI resolution limit is 100 nm or smaller.

Comment 8. “The simulated diffraction limit is about 3.2 pixels.”: please describe briefly how this is calculated

Reply: We have revised this sentence as below to explain it.

The PSF is generated with a low-pass filter whose cut-off frequency is 1/3.2 times of the Nyquist frequency. Therefore, the simulated diffraction limit is about 3.2 pixels.

Comment 9. “an intermittent emitter with a size of about 1.34 μm (Supplementary Fig. S1a)”: please improve the description and the quality of the information provided in supp.fig.1.

Reply: Yes, we revised this part of supplementary thoroughly according to your several comments. The figure’s quality is improved. We have revised the existing paragraph and added two more paragraphs to describe the experiments in detail. The supplementary Fig. S1 and its description text are as follows.

The optical setup for our experimental demonstration of stochastic optical scattering localization imaging (SOSLI) is depicted schematically in supplementary Fig. S1. It consists of two parts: the object simulator and the imaging setup. The former is designed for convenient generation of various objects with blinking point sources. We replace the projection lens of a commercial projector (Acer X113PH) by a microscope objective (40x, numerical aperture: NA=0.65) to de-magnify projector pixels to squares of $1.34 \times 1.34 \mu\text{m}^2$ at the object plane. There is an iris placed at the projected plane of the projector to remove its stray light. And the second iris is at the object plane to further block unwanted light from the projector and environment. Light from the object passing through both scattering media and the imaging iris is captured by a camera sensor (Andor Neo 5.5, 2560×2160 pixels, and 6.5- μm pixel size). The scattering media are a ground glass diffuser (a static one) or a fresh chicken eggshell membrane (a dynamic one) in our demonstration. An optical filter (Thorlabs FB550-10, 550 nm wavelength, and 10 nm full-width at half-maximum) is mounted on the camera to narrow the optical spectrum. Blinking point sources are generated by randomly blinking projector pixels.

For invasive measurement of the point spread function (PSF), only one center pixel is turned on. The exposure time to capture each speckle pattern is 5-10 seconds to gain good signals.

Supplementary Fig. S1. Optical setup to demonstrate SOSLI for non-invasive super-resolution imaging through strongly scattering media. (a) Object-simulator, which is designed for generating various microscopic objects at the object plane. **(b)** Simple optical configuration for imaging setup where $u = 10$ mm and $v = 100$ mm.

Although the projector's projection lens is removed so that it does not project to a huge screen, there are still multiple unknown optical components inside as the projector still projects images to the plane of the first iris. Because of these unknown optics, we cannot calculate the demagnification factor from the projector to the object plane. We measure the pixel size of $1.34 \mu\text{m}$ by a mechanical approach. We turn on 200 pixels to make a few-hundred-micrometer bright line. A blade is then moved along the line to block a portion of it and the transmitted light intensity is monitored by a power meter. We used a differential actuator to control the blade position with sub-micrometer precision. By coordinating the transmission intensity with the blade position, the line length and then the pixel size can be calculated.

Comment 10. How the distances “ u ” and “ v ” of supp. fig. 1 have been chosen?

Reply: “ u ” and “ v ” for our experiments were chosen to meet the criteria as follows.

- “ u ” is far enough so that the entire object is within the memory effect region of scattering media. But “ u ” should not be too far as it will significantly reduce the signal.
- v/u is large enough to achieve significant magnification so that camera pixels of $6.5 \mu\text{m}$ can resolve the smallest feature of objects.

In supplementary material, we have added a paragraph in the section: Optical experiment setup. This paragraph describes the simple procedure to get the experimental setup.

The optical configuration for the imaging side is relatively simple. To get easy success, we choose the distances u and v to achieve adequate magnification while maintaining acceptable signal and sufficient memory effect. We first select the smallest u so that our phase retrieval algorithm is still successful; this confirms our sample is within the memory effect of the

scattering media. Then the distance v is then chosen so that the magnification ($M = v/u$) is significantly large to resolve objects at high resolution by camera pixels. In non-invasive applications, neither u nor the memory effect region is our choice, but both determine the maximum size of measurable objects. Therefore, we simply need to put the camera and adjust the camera position to capture the signal for SOSLI. A smaller v will gain the SNR at the cost of reduced magnification and vice versa.

Comment 11. "We now utilize SOSLI to localize and then superpose emitters in a set of first 50 stochastic patterns": these approach and the numbers of pattern before the evaluation of the PF depends on a prior knowledge of the decorrelation characteristics of the scattering layer, which can be unknowns in most of the applications: I think is needed a discussion of this point.

Reply: This is a great comment. Thank the reviewer for this point. We have added a paragraph after the most adaptive SOSLI to address this point (page 13). For your convenience, it is read as below.

We would like to note that the deconvolution and localization algorithms are faster than the phase retrieval algorithm by orders of magnitude. If we want to save the post-processing time, we should utilize the adaptive SOSLI with the most acceptable number of speckle patterns in a group. In many applications, the decorrelation time is unknown, the number of speckle patterns per group needs to be decided non-invasively by calculating the correlation of several estimated PSFs from individual speckle patterns at different time intervals. If post-processing time is not a concern, we can always use the most adaptive SOSLI approach for all the scattering media, whose correlation in two consecutive imaging shots is not smaller than 0.2.

Comment 12. "The SNR is very low with SOSLI and we cannot easily increase it simply by increasing the integration time because we need stochastic nature of blinking emitters from frame to frame": if the authors claim is not possible to have higher SNR with such a simple proof of principle setup, which are the possibilities to move to "real" applications? please discuss this point

Reply: In fact, our writing for discussion is beyond the current proof of principle setup. We discuss the real applications with individual blinking emitters in this paragraph. In short, for real applications, we will have to rely on inorganic quantum emitters with very high quantum yield (for brightness) and suppressed blinking rate³⁹ (for increasing exposure time). For current proof of principle setup, we simply increase the exposure time for each frame to 5-10 seconds and get good SNR. And certainly, we can increase more to get higher SNR.

We think the transition from proof-of-principle setup to the real applications was abrupted in writing and led to this misunderstanding of "not possible to have higher SNR with our proof-principle setup". We revised this sentence as follows.

The SNR will be very low in SOSLI for applications with blinking emitters because SOSLI needs only one point source to be active within the area of diffraction limit in each frame. We might not improve SNR simply by increasing the frame's integration time as there will be more chances for multiple point sources to be active within diffraction limit area.

The full discussion on real applications is then followed in the same paragraph with supported results from our simulation for experiments with low-photon-budget emitters. The discussion is as follows.

Besides the need for a low noise and high quantum efficiency camera to enhance SNR, high photon budget (number of photons per blink per emitter) is a challenging requirement. Ultrabright single fluorophores or quantum emitters could emit at the best $10^4 - 10^6$ photons per blink to a camera³⁸. With 10-1000 pixels in a diffraction limited region for standard imaging with an objective lens and transparent samples, this is equivalent to $10^1 - 10^3$ photons per useful pixel (PPP). However, for scattering media, these photons are spread in million pixels, this implies much less than $10^{-2} - 10^0$ photons per pixel (PPP), lower than the camera's noise floor. The low photon budget also implies the uncertainty of the captured speckle patterns regardless of noise (supplementary Fig. S10). Such quantum uncertainty is the significant noise source for SOSLI with intermittent emitters. Our simulation (supplementary Fig. S11) for static scattering media and noiseless experiments shows that 11% of success rate for phase retrieval and localization can be achieved with about 1 PPP. After PSF is estimated, the success rate for deconvolution and localization is very high such as 68% for 0.1 PPP and 100% for 0.45 PPP and above (supplementary Fig. S12). Understandably, our bottleneck is the phase retrieval algorithm; and multi-million photons are required for SOSLI. The future development of inorganic quantum emitters with very high quantum yield (for brightness) and suppressed blinking rate³⁹ (for a longer exposure time of each stochastic image) could enable SOSLI in practical applications.

Comment 13. Data sharing link was not available while revising the manuscript

Reply: We are sorry for this. We should have noted to editor and reviewers. We disabled the link as we saw people quickly downloaded it after we enrolled in www.researchsquare.com during submission to Nature Comm.

The link to shared data is here: <https://figshare.com/s/8bbac280c398d3ca73ca>. It is a confidential link currently. It will be published together with our paper.

Some minor points:

Comment 14. I suggest to improve the overall quality of Supp.Fig.1: the first two quotes from the left does not have text, I would improve consistency of perspective, resolution of the images (or move from raster to vector graphics), make more clear the light propagation (e.g. a kind of raytrace could be to my view more effective).

Reply: We would like to thank the reviewer for multiple comments to improve Figure S1 and its discussion in the text. The figure is now adjusted as below with high quality as we exported it from vector graphic in Adobe Illustrator into .tif format (lossless imaging format). And its discussion is with much more details about the experimental setup, the characterization of pixel size and procedure for optimizing the distances. The entire supplementary section 1: Optical experiment setup has been transformed as below.

1. Optical experiment setup

The optical setup for our experimental demonstration of stochastic optical scattering localization imaging (SOSLI) is depicted schematically in supplementary Fig. S1. It consists of two parts: the object simulator and the imaging setup. The former is designed for convenient generation of various objects with blinking point sources. We replace the projection lens of a commercial projector (Acer X113PH) by a microscope objective (40x, numerical aperture: $NA=0.65$) to de-magnify projector pixels to squares of $1.34 \times 1.34 \mu\text{m}^2$ at the object plane. There is an iris placed at the projected plane of the projector to remove its stray light. And the second iris is at the object plane to further block unwanted light from the projector and environment. Light from the object passing through both scattering media and the imaging iris is captured by a camera sensor (Andor Neo 5.5, 2560×2160 pixels, and $6.5\text{-}\mu\text{m}$ pixel size). The scattering media are a ground glass diffuser (a static one) or a fresh chicken eggshell membrane (a dynamic one) in our demonstration. An optical filter (Thorlabs FB550-10, 550 nm wavelength, and 10 nm full-width at half-maximum) is mounted on the camera to narrow the optical spectrum. Blinking point sources are generated by randomly blinking projector pixels. For invasive measurement of the point spread function (PSF), only one center pixel is turned on. The exposure time to capture each speckle pattern is 5-10 seconds to gain good signals.

Supplementary Fig. S1. Optical setup to demonstrate SOSLI for non-invasive super-resolution imaging through strongly scattering media. (a) Object-simulator, which is designed for generating various microscopic objects at the object plane. **(b)** Simple optical configuration for imaging setup where $u = 10\text{ mm}$ and $v = 100\text{ mm}$.

Although the projector's projection lens is removed so that it does not project to a huge screen, there are still multiple unknown optical components inside as the projector still projects images to the plane of the first iris. Because of these unknown optics, we cannot calculate the demagnification factor from the projector to the object plane. We measure the pixel size of $1.34\text{ }\mu\text{m}$ by a mechanical approach. We turn on 200 pixels to make a few-hundred-micrometer bright line. A blade is then moved along the line to block a portion of it

and the transmitted light intensity is monitored by a power meter. We used a differential actuator to control the blade position with sub-micrometer precision. By coordinating the transmission intensity with the blade position, the line length and then the pixel size can be calculated.

The optical configuration for the imaging side is relatively simple. To get easy success, we choose the distances u and v to achieve adequate magnification while maintaining acceptable signal and sufficient memory effect. We first select the smallest u so that our phase retrieval algorithm is still successful; this confirms our sample is within the memory effect of the scattering media. Then the distance v is then chosen so that the magnification ($M = v/u$) is significantly large to resolve objects at high resolution by camera pixels. In non-invasive applications, neither u nor the memory effect region is our choice, but both determine the maximum size of measurable objects. Therefore, we simply need to put the camera and adjust the camera position to capture the signal for SOSLI. A smaller v will gain the SNR at the cost of reduced magnification and vice versa.

Comment 15. "transfer matrix measurement" : i think in the field is more often used the term "transmission matrix" instead of "transfer matrix". Unless there is a clear reason to use "transfer", I suggest changing it to be more clear.

Reply: Thank reviewer. We agree with this point and change it accordingly.

Comment 16. Another

17. "approach relies on memory effect of light through scattering media and deconvolution imaging": while the phenomenon exploited by the former part of the sentence is memory effect, in the latter the authors refer to the "deconvolution imaging", which now sounds more a technique, not anymore a effect to exploit, (which moreover I assume exploit as well the memory effect/isoplanatism/shift invariance). I think the sentence can bring confusion to the reader, so I suggest to revise it.

Reply: We thank the reviewer for this important point. And you are correct that memory effect implies both isoplanatism and shift invariance in this context. We revise this part as follows.

Another approach relies on the memory effect of light through scattering media^{24,25}, which implies a shift-invariant point spreading function (PSF). A scattering medium with a known PSF (often being measured invasively) can be treated as a scattering lens for imaging by deconvolution²⁶⁻²⁸ (Fig. 1b).

Comment 18. Report (in response to the referees) as reference of egg shell thickness the site of a mayonnaise producer, without even a datasheet nor the date when the site has been visited, must be not considered a valid response. I suggest finding a better and more scientifically relevant reference.

Reply: We agree with you on this point. We now cite a good reference in the supplementary material as:

The thickness L of a printing paper sheet¹ and a chicken eggshell membrane² is about 89 μm and 70 μm , respectively; we can calculate their transport MFP of 12.5 μm and 19.2 μm , respectively.

[1] Badon, A. et al. Smart optical coherence tomography for ultra-deep imaging through highly scattering media. Science Advances 2, e1600370, (2016).

[2] Stadelman, W. J. & Cotterill, O. J. Egg science and technology. (Avi Pub. Co., 1986).

Comment: Although the proposed technique exploits well known techniques, the article describes a simple and effective approach, which has a degree of novelty. Moreover the validation has been with a simple and intuitive setup, which to my view must be considered a plus. However the article still needs to be extensively revised, in order to be clear and the results consistent and corroborated by good statistics. While it could be ready with some minor revision for a more specific journal, I think the quality of the manuscript as presented does not deserve the publication in an high impact and broad audience journal, such as Nature Communication.

Reply: Thank you for your encouraging words on our technique. We have revised the manuscript thoroughly, engaged an English native scientist outside the field to work on language. The revised version is in much better shape with clear messages. We also attached the tracked change version of our manuscript for reviewers and editor to see our revision.

In the last version, the results were consistent, but our language together with mixing statistic concepts made some confusion for readers. Thanks to your suggestions, we have fixed these confusions. We hope you agree with us that our manuscript is attractive for a broad audience of Nat. Comm. journal. Especially with available Matlab codes (published with paper later) and the simplicity, intuitiveness of optical setup, most researchers in imaging research can repeat our experiments and reveal “surprises” behind scattering media.

Reviewers' Comments:

Reviewer #2:

Remarks to the Author:

The authors revised the manuscript appropriately. I agree to publish this paper in Nature Communications.

Reviewer #3:

Remarks to the Author:

I am positively surprised by the quality of the new manuscript and the responses. As described by the authors, I think that now the "manuscript is attractive for a broad audience of Nat.Comm. journal. Especially with available Matlab codes and the simplicity, intuitiveness of optical setup, most researchers in imaging research can repeat our experiments and reveal "surprises" behind scattering media."

I hope the last round of revision helped the authors to improve the manuscript without losing their initial vision of the paper.

I suggest the publication of the manuscript by Nat.Comm. upon the answer by the authors to a major point, and eventually to some minors:

Major:

In the manuscript I read "We replace the projection lens of a commercial projector (Acer X113PH) by a microscope objective (40x, numerical aperture: NA=0.65)", that means that the projection optics has only an iris between the DMD and the microscope objective? In that case I assume the objective cannot be an infinity corrected one, and so you should know exactly the demagnification (40x), which should lead to a demagnification of the dmd pixels between 0.135um and 0.34um (assuming standard devices from Texas Instruments, with mirrors between 5.4um and 13.6um). However, if the configuration is an infinity corrected one, with the tube lens replaced by part of the DMD's projection optics, the sentences across the text, and in particular the one noted above, should be more clear (e.g. why the projection lens has been replaced? which is the tube lens working with the objective used to create the image?). I strongly suggest also to report explicitly also the exact brand and model of the objectives. I think a detailed report of the material used in the setup would largely improve the reproducibility of the measurements (as there are other parameters than magnification and numerical aperture which can affect the results).

Minors:

Fig.1.a : "membrane sandwiched between two coverglasses" -> "membrane sandwiched between two coverglasses"

To protect the work done by the authors, I suggest to update the license of the code as suggested by "code and software submission checklist" proposed by the editor, so using a licence approved by the open source initiative

In the text "Egg science and technology" I read "Two membranes (consisting of protein fibers) reside between albumen and the inner surface of the shell. The total thickness of the two membranes have been found to range from 73 to 114um in eggs from White Leghorn and New Hampshire pullets (Tung and Richards 1972)." [p.100]. So 70um seems to be a lower limit for the membrane thickness, not the average.

In the text "Egg science and technology" I don't find any reference to the sentence "The chicken eggshell membrane has similar effective thickness when fresh; then this increases when dried."; moreover, to the reader seems counterintuitive that as the membrane dry it get thicker. Please clarify the sentence. Moreover I suggest to add the thickness also in that sentence, and so, in the methods.

"more than 0.65 (shaded area in Fig. 5a), while the actual correlation reduces to 0.2" I would place 0.20 instead of 0.2, meaning that the correlation measurements have the same precision. Similarly I suggest to add zero across the whole manuscript when zero is significant.

Our responses to reviewers' comments

Reviewer #2 (Remarks to the Author):

The authors revised the manuscript appropriately. I agree to publish this paper in Nature Communications.

Response: We are glad to hear this. We thank you very much for your thorough review, which improves our work significantly.

Reviewer #3 (Remarks to the Author):

I am positively surprised by the quality of the new manuscript and the responses. As described by the authors, I think that now the "manuscript is attractive for a broad audience of Nat.Comm. journal. Especially with available Matlab codes and the simplicity, intuitiveness of optical setup, most researchers in imaging research can repeat our experiments and reveal "surprises" behind scattering media."

I hope the last round of revision helped the authors to improve the manuscript without losing their initial vision of the paper.

I suggest the publication of the manuscript by Nat.Comm. upon the answer by the authors to a major point, and eventually to some minors:

Response: Indeed, the last round of revision was a significant enhancement of our work. We thank both reviewers for your thorough reports. We thank you very much for your kind words this round. We are glad to respond to your comments point-by-point as below.

Major:

Comment: In the manuscript I read "We replace the projection lens of a commercial projector (Acer X113PH) by a microscope objective (40x, numerical aperture: NA=0.65)", that means that the projection optics has only an iris between the DMD and the microscope objective? In that case I assume the objective cannot be an infinity corrected one, and so you should know exactly the de-magnification (40x), which should lead to a de-magnification of the dmd pixels between 0.135um and 0.34um (assuming standard devices from Texas Instruments, with mirrors between 5.4um and 13.6um). However, if the configuration is an infinity corrected one, with the tube lens replaced by part of the DMD's projection optics, the sentences across the text, and in particular the one noted above, should be more clear (e.g. why the projection lens has been replaced? which is the tube lens working with the objective used to create the image?). I strongly suggest also to report explicitly also the exact brand and model of the objectives. I think a detailed report of the material used in the setup would largely improve the reproducibility of the measurements (as there are other parameters than magnification and numerical aperture which can affect the results).

Response: The commercial projector (Acer X113PH) has other lenses/optics inside for various functions such as zooming, skewing, keystone correction, etc. We only remove the biggest lens (the outermost lens), which is the projection lens to project a huge screen (certainly, we do not want this). After removing the outermost lens, we see that the projector still project a small screen in the

front. We placed the first iris at that small screen to remove stray light significantly. We can see that the remaining optics inside Acer X113PH plays the role of a relay lens in our setup (not a tube lens). Then our microscope objective simply de-magnifies the small screen to an even smaller size for our experiments. The objective lens is very old in the lab, not an infinity-corrected one. We cannot identify the brand (Fig. R1). In fact, we only use it as a short focal-length lens to de-magnify the projection image without losing much light. We only put the objective lens about 5 cm (instead of 16 cm) after the first iris and observe the final image after the objective. Hence, the actual de-magnification is not 40X as objective design. One can set up as the objective design; however, the large distance from the first iris to the objective lens will significantly reduce light intensity. We also need to consider the diffraction limit for this projection system, i.e. we cannot expect to achieve 0.135 μ m and 0.34 μ m pixel size. We would like to emphasize that there is no special objective lens, and we believe one can use a short-focal length lens to achieve the same goal.

Fig. R1: Microscope objective to de-magnify the projection image in the object simulation setup.

We certainly can find out the de-magnification factor by disassembling the projector to understand its optics, then with some guess of Texas Instrument's DMD pixel size, we can estimate our de-magnified pixel size. However, the error will be large compared to our mechanical approach as described in the supplementary information.

We understand the reviewer's concern about our experiment's reproducibility related to our description's clarity in the experimental setup. We revised the supplementary section 1 carefully to make sure the other researchers can repeat our experiments as below.

Revision:

"The optical setup for our experimental demonstration of stochastic optical scattering localization imaging (SOSLI) is depicted schematically in supplementary Fig. S1. It consists of two parts: the object simulator and the imaging setup. The former is designed for convenient generation of various objects with blinking point sources. To create microscopic objects, we certainly do not want the projector's projection lens as it magnifies images to a

huge screen. We replace the projection lens of a commercial projector (Acer X113PH) by a microscope objective (40x, numerical aperture: NA=0.65, not an infinity-corrected objective) to de-magnify projector pixels to squares of $1.34 \times 1.34 \mu\text{m}^2$ at the object plane. There is an iris placed at the projected plane of the projector to remove its stray light. And the second iris is at the object plane to further block unwanted light from the projector and environment. Light from the object passing through both scattering media and the imaging iris is captured by a camera sensor (Andor Neo 5.5, 2560×2160 pixels, and 6.5- μm pixel size). The scattering media are a ground glass diffuser (a static one) or a fresh chicken eggshell membrane (a dynamic one) in our demonstration. An optical filter (Thorlabs FB550-10, 550 nm wavelength, and 10 nm full-width at half-maximum) is mounted on the camera to narrow the optical spectrum. Blinking point sources are generated by randomly blinking projector pixels. For invasive measurement of the point spread function (PSF), only one center pixel is turned on. The exposure time to capture each speckle pattern is 5-10 seconds to gain good signals.

Although the projector's projection lens is removed so that it does not project to a huge screen, there are still multiple unknown optical components inside for various projector functions such as zooming, keystone correction, etc. In fact, ~~as~~ the projector still projects images to the plane of the first iris after projection lens removal. Hence, the projector's remaining optics plays as a relay lens from the digital micromirror device (DMD) to the first iris plane. The objective lens is placed just 5 centimeters from the first iris plane to de-magnify the image further. We simply use the objective lens as a short-focal-length lens rather than set it up to achieve 40X de-magnification. One can do the same with a normal short-focal-length lens. We would like to note that there is also a diffraction limit for the projection image; and more de-magnification requires further distance from the objective to the first iris plane, leading to reduced signal.

~~Because~~ Due to these unknown remaining relay optics in the projector, we cannot calculate the de-magnification factor from the projector-DMD to the object plane. We measure the pixel size of 1.34 μm by a mechanical approach. We turn on 200 pixels to make a few-hundred-micrometer bright line. A blade is then moved along the line to block a portion of it and the transmitted light intensity is monitored by a power meter. We used a differential actuator to control the blade position with sub-micrometer precision. By coordinating the transmission intensity with the blade position, the line length and then the pixel size can be calculated."

Minors:

Comment: Fig.1.a : "membrane sandwiched between two coverglasses" -> "membrane sandwiched between two coverglasses"

Response: We appreciate your help to improve our work. We have corrected this one in the revised version.

Comment: To protect the work done by the authors, I suggest to update the license of the code as suggested by "code and software submission checklist" proposed by the editor, so using a licence approved by the open source initiative

Response: We thank you for your suggestion. We have updated the license of the code with "The MIT License". The content of our license.txt is as follows.

The MIT License (MIT)

Copyright (c) 2020 Cuong Dang Group

Permission is hereby granted, free of charge, to any person obtaining a copy of this software and associated documentation files (the "Software"), to deal in the Software without restriction, including without limitation the rights to use, copy, modify, merge, publish, distribute, sublicense, and/or sell copies of the Software, and to permit persons to whom the Software is furnished to do so, subject to the following conditions:

The above copyright notice and this permission notice shall be included in all copies or substantial portions of the Software.

THE SOFTWARE IS PROVIDED "AS IS", WITHOUT WARRANTY OF ANY KIND, EXPRESS OR IMPLIED, INCLUDING BUT NOT LIMITED TO THE WARRANTIES OF MERCHANTABILITY, FITNESS FOR A PARTICULAR PURPOSE AND NONINFRINGEMENT. IN NO EVENT SHALL THE AUTHORS OR COPYRIGHT HOLDERS BE LIABLE FOR ANY CLAIM, DAMAGES OR OTHER LIABILITY, WHETHER IN AN ACTION OF CONTRACT, TORT OR OTHERWISE, ARISING FROM, OUT OF OR IN CONNECTION WITH THE SOFTWARE OR THE USE OR OTHER DEALINGS IN THE SOFTWARE

Comment: In the text "Egg science and technology" I read "Two membranes (consisting of protein fibers) reside between albumen and the inner surface of the shell. The total thickness of the two membranes have been found to range from 73 to 114um in eggs from White Leghorn and New Hampshire pullets (Tung and Richards 1972)." [p.100]. So 70um seems to be a lower limit for the membrane thickness, not the average.

Response: We thank you very much for your thorough review. This was our early mistake in response to Nature Photonics' reviewer as we got the number from a website which cites this reference. We are sorry for this. We revised our manuscript and chose the thickness of 90 μm for both paper and membrane. The revision as follows.

Revision:

“The thickness L of both a printing paper sheet¹ and a chicken eggshell membrane² is about ~~89-90 μm and 70 μm , respectively~~; we can calculate their transport MFP of ~~12.516 μm and 19.225 μm , respectively.~~”

In the text "Egg science and technology" I don't find any reference to the sentence "The chicken eggshell membrane has similar effective thickness when fresh; then this increases when dried."; moreover, to the reader seems counterintuitive that as the membrane dry it get thicker. Please clarify the sentence. Moreover I suggest to add the thickness also in that sentence, and so, in the methods.

Response: We implied "the similar effective thickness" compared to scattering media (optical diffuser). The sentence is confusing. We are sorry for this.

Our effective thickness conclusion is merely based on the measurement of the memory effect region. However, this might lead to misunderstanding about the thickness of ground glass with the thickness of membrane. We also realize that the ground glass diffuser and eggshell membrane are very different scattering media in nature. The former is simply a rough surface between two different refractive index environments while the latter is volumetric scattering media. Conclusion about their thickness from memory effect region measurement might be misleading. Also, when membrane dries, the properties of scattering "elements" could change and lead to reduce memory effect, which is interpreted as increased effective scattering thickness and not necessarily physical thickness. We have changed the sentences in the method to discuss just the memory effect as below. Then the membrane thickness information is added in the main text (page 15). We also added reference 40 into the main text.

Revision:

“The static scattering media are 120 grit ground glass diffusers from Thorlabs or Edmund. The memory effect region is 15 mrad; ~~which implies the effective thickness of the scattering media is 18 μm~~ (Ref. 27). The fresh chicken eggshell membrane has a similar effective thickness memory effect region as that of the glass diffuser when fresh; then this increases the memory effect region decreases when membrane dries.”

On page 15:

“The chicken eggshell membranes (thickness of about 90 μm)⁴⁰ with 3D volumetric scatterers are very strong scattering media whose ratio R is 7.3 and 3.6 for scattering and transport MFP (l_s and l_t), respectively (more detail in supplementary Fig. S13-S15).”

Added Reference:

40. Stadelman, W. J. & Cotterill, O. J. *Egg science and technology*. (Avi Pub. Co., 1986).

Comment: "more than 0.65 (shaded area in Fig. 5a), while the actual correlation reduces to 0.2" I would place 0.20 instead of 0.2, meaning that the correlation measurements have the same precision. Similarly I suggest to add zero across the whole manuscript when zero is significant.

Response: We thank the reviewer for this point, we have adjusted them accordingly.